# First-in-Class Humanized Antibody against Alternatively Spliced Tissue Factor Augments Anti-Metastatic Efficacy of Chemotherapy in a Preclinical Model of Pancreatic Ductal Adenocarcinoma

**DOI:** 10.3390/ijms25052580

**Published:** 2024-02-23

**Authors:** Clayton S. Lewis, Charles Backman, Sabahat Ahsan, Ashley Cliff, Arthi Hariharan, Jen Jen Yeh, Xiang Zhang, Changchun Xie, Davendra P. S. Sohal, Vladimir Y. Bogdanov

**Affiliations:** 1Division of Hematology/Oncology, Department of Internal Medicine, College of Medicine, University of Cincinnati, Cincinnati, OH 45267, USA; clayton.lewis@uc.edu (C.S.L.); backmaca@mail.uc.edu (C.B.); ahsansh@mail.uc.edu (S.A.); sohalda@uc.edu (D.P.S.S.); 2Lineberger Comprehensive Cancer Center, University of North Carolina at Chapel Hill, Chapel Hill, NC 27599, USA; ashley_cliff@med.unc.edu (A.C.); arthih@unc.edu (A.H.); jen_jen_yeh@med.unc.edu (J.J.Y.); 3Departments of Surgery and Pharmacology, University of North Carolina at Chapel Hill, Chapel Hill, NC 27599, USA; 4Division of Environmental Genetics and Molecular Toxicology, Department of Environmental and Public Health Sciences, College of Medicine, University of Cincinnati, Cincinnati, OH 45267, USA; xiang.zhang@uc.edu; 5Division of Biostatistics and Bioinformatics, Department of Environmental and Public Health Sciences, College of Medicine, University of Cincinnati, Cincinnati, OH 45267, USA; changchun.xie@uc.edu

**Keywords:** pancreatic ductal adenocarcinoma (PDAC), alternatively spliced tissue factor (asTF), humanized monoclonal antibody

## Abstract

Alternatively spliced tissue factor (asTF) promotes the progression of pancreatic ductal adenocarcinoma (PDAC) by activating β1-integrins on PDAC cell surfaces. hRabMab1, a first-in-class humanized inhibitory anti-asTF antibody we recently developed, can suppress PDAC primary tumor growth as a single agent. Whether hRabMab1 has the potential to suppress metastases in PDAC is unknown. Following in vivo screening of three asTF-proficient human PDAC cell lines, we chose to make use of KRAS G12V-mutant human PDAC cell line PaCa-44, which yields aggressive primary orthotopic tumors with spontaneous spread to PDAC-relevant anatomical sites, along with concomitant severe leukocytosis. The experimental design featured orthotopic tumors formed by luciferase labeled PaCa-44 cells; administration of hRabMab1 alone or in combination with gemcitabine/paclitaxel (gem/PTX); and the assessment of the treatment outcomes on the primary tumor tissue as well as systemic spread. When administered alone, hRabMab1 exhibited poor penetration of tumor tissue; however, hRabMab1 was abundant in tumor tissue when co-administered with gem/PTX, which resulted in a significant decrease in tumor cell proliferation; leukocyte infiltration; and neovascularization. Gem/PTX alone reduced primary tumor volume, but not metastatic spread; only the combination of hRabMab1 and gem/PTX significantly reduced metastatic spread. RNA-seq analysis of primary tumors showed that the addition of hRabMab1 to gem/PTX enhanced the downregulation of tubulin binding and microtubule motor activity. In the liver, hRabMab1 reduced liver metastasis as a single agent. Only the combination of hRabMab1 and gem/PTX eliminated tumor cell-induced leukocytosis. We here demonstrate for the first time that hRabMab1 may help suppress metastasis in PDAC. hRabMab1’s ability to improve the efficacy of chemotherapy is significant and warrants further investigation.

## 1. Introduction

PDAC is associated with high rates of venous thromboembolism (VTE); one of the key contributors to this morbidity is tissue factor (TF, also known as CD142, thromboplastin, coagulation factor III) [1]. The much-studied, plasma membrane-bound form of TF protein termed full-length (fl)TF, is the obligatory cofactor of the plasma serine protease fVIIa and triggers blood clotting either upon tissue damage, or aberrant expression in cells that come in contact with circulating blood; in PDAC, cancer cell-associated and extracellular vesicle-bound flTF both contribute to VTE [2]. Hypoxia synergizes with such oncogenic drivers as KRAS to induce TF (*F3*) gene expression via the amplification of PI3K-Akt and p38-NFkB signaling pathways, both of which are prominent in KRAS-mutant cancers including PDAC [3]; HIF-1α can also induce TF in cancer cells indirectly, via the upregulation of VEGF expression [4]. In addition to cancer cells, stromal cells such as monocytes/macrophages, fibroblasts, and microvascular endothelial cells express TF in cancer lesions [5]. Aside from causing thrombosis in PDAC, high TF expression was long known to correlate with PDAC’s histological grade [6]; in 1999, it was reported that TF can promote PDAC growth and tumor cell invasion in vivo [7]. Nitori and colleagues suggested that TF may have prognostic significance in PDAC: “high TF” patients presented with larger tumors and more advanced metastatic disease with TF prominently expressed at the invasive front of the primary tumor [8]. More recently, the Flick laboratory reported that the flTF/fVIIa complex can contribute to metastatic seeding and immune evasion by cleaving protease-activated receptors on PDAC cell surfaces [9], and this year, Zhang et al. reported that TF overexpression can promote resistance to the newest class of KRAS-G12C inhibitors [10].

Another layer of complexity to TF function involves activity that is not protease dependent, but rather integrin mediated; it is largely executed by TF’s minimally coagulant alternatively spliced form, asTF. Unlike flTF, asTF lacks a transmembrane domain and can, thus, be secreted as a free protein; a splicing-dependent shift in asTF’s open reading frame creates a unique 40 amino acid C-terminal epitope in asTF, which makes it possible to develop asTF-specific antibodies [11]. asTF binds a subset of β1 integrins in close proximity to the “knee” region, causing a conformational change that amplifies integrin-linked outside-in signaling. When bound to integrins on benign cells, e.g., endothelium, asTF promotes cell migration and the expression of leukocyte adhesion molecules, yet not cell proliferation [12]; however, when bound to integrins on malignant cells, asTF fuels both proliferation as well as systemic spread [13,14]. Given asTF’s cell-agonist properties, along with its dispensability for normal hemostasis, asTF is an attractive therapeutic target.

In 2021, we reported the results of the first study that evaluated the in vivo efficacy of an asTF-specific, inhibitory, humanized antibody termed hRabMab1 [15]. We found that hRabMab1 was able to suppress the growth of pre-formed, orthotopically grown PDAC tumors (KRAS G12D-mutant cell line Pt45.P1) when administered intravenously as a single agent. hRabMab1 exhibited a favorable pharmacokinetic (PK) profile in mice with no toxicity detected at the dose of 18 mg/kg; examination of PDAC tumor tissue post-hRabMab1 treatment showed the reduction of cancer cell proliferation and decreased monocyte/macrophage infiltration of the lesions. In this study, we assessed hRabMab1’s ability to suppress the progression of experimental PDAC in a model featuring a more aggressive asTF-proficient human PDAC cell line—KRAS G12V-mutant PaCa-44 cells [16]—alone and in combination with a standard-of-care regimen, gemcitabine/paclitaxel (gem/PTX).

## 2. Results

### 2.1. Orthotopic Implantation of PaCa44 Cells Yields Stroma-Rich Primary Tumors with Spontaneous Metastases

Recent studies have demonstrated that circulating tumor cells (CTCs) arising endogenously from solid primary tumors undergo a multi-step metastatic process that is not recapitulated by cell lines grown in vitro limiting the value of tumor cell-seeding approaches such as tail vein injections and hemi-splenic injections for studying PDAC metastatic seeding in the lung and the liver, respectively [17,18]. Given our desire to determine the anti-metastatic potential of hRabMab1, we first sought to ascertain the metastatic potential of three asTF-proficient human PDAC cell lines grown orthotopically in SCID mice. The expression profile of asTF-target integrins and the ability of each cell line to yield spontaneous metastases in an orthotopic setting are shown in Figure 1.

asTF-proficient, KRAS G12V-mutant cell line PaCa-44 yields reproducible-size, aggressive primary orthotopic tumors that spread spontaneously to PDAC-relevant anatomical sites (Figure 2); other useful features of the PaCa-44 model comprise its 100% penetrance of metastases to the site of surgical incision (wound closure area in the abdominal wall), and severe leukocytosis.

### 2.2. Effects of hRabMab1 in the Primary Tumor Tissue

We determined that sequential administration of gem/PTX (50 mg/kg/3 mg/kg, respectively) in line with Wolfe et al. [19] resulted in a significant reduction in primary volume (~80%, *p* = 0.002, vehicle vs. gem/PTX). To study the effect of hRabMab1 in this model, 5 × 10^5^ luciferase-labeled PaCa-44 cells were implanted into the pancreata of NOD.scid mice (*n* = 40); 10 days post-implantation, mice were randomized into 4 cohorts: vehicle; hRabMab1 (IV at 18 mg/kg); gem/PTX (50 mg/kg and 3 mg/kg, respectively); and the combination of hRabMab1 with gem/PTX. Post-mortem analysis of primary tumors revealed that hRabMab1 did not significantly impact the primary tumor volume when used alone or in combination with gem/PTX (Figure 3).

Immunohistochemical (IHC) analysis revealed that, unlike in thin-capsule forming Pt45.P1 tumors, which are penetrable by hRabMab1 as a single agent [15], intratumoral hIgG was not detectable in well-encapsulated PaCa-44 tumors in mice that received hRabMab1 as a single agent. However, when hRabMab1 was combined with gem/PTX, intratumoral hIgG was found to be abundant throughout the tumor tissue (Figure 4A). Ki67+ signal was significantly suppressed by the addition of hRabMab1 to gem/PTX; there were also significantly fewer neutrophils, monocytes, and microvessels in hRabMab1+gem/PTX tumors compared to gem/PTX tumors (Figure 4B–E and Appendix A).

RNA-seq analysis of the tumor tissue followed by gene-set enrichment of differentially expressed genes (three tumors per cohort: control; gem/PTX; and hRabMab1+gem/PTX, Figure 5) revealed that the addition of hRabMab1 to gem/PTX downregulated tubulin binding and microtubule motor activity. Genes involved in neovascularization were upregulated in response to gem/PTX and the addition of hRabMab1 to gem/PTX weakened this compensatory effect, which is consistent with our IHC data (Figure 4E).

### 2.3. Systemic Effects of hRabMab1

As assessed by quantitative luciferase imaging, cumulative metastatic spread (liver, lung, abdominal cavity) was not significantly reduced by gem/PTX alone; however, the addition of hRabMab1 to gem/PTX significantly reduced whole-body metastatic spread (*p* = 0.0415, vehicle vs. hRabMab1+gem/PTX, Figure 6).

In the liver, hRabMab1 significantly reduced metastatic burden as a single agent to a comparable degree to that achieved by gem/PTX (hRabMab1: *p* = 0.0089 vs. vehicle; gem/PTX: *p* = 0.0008 vs. vehicle; Appendix A). At the surgical incision site, where PaCa-44 metastases routinely engraft, the following results were obtained: vehicle, 100% penetrance (10/10); hRabMab1, 100% penetrance (8/8); gem/PTX, 44% penetrance (4/9); hRabMab1+gem/PTX, 0% penetrance (0/8; *p* = 0.0378 for gem/PTX vs. hRabMab1+gem/PTX, Chi-square test with 0.1 replacing 0). Only the combination of hRabMab1 with gem/PTX was able to eliminate neutrophil-driven leukocytosis in our model (Figure 7 and Appendix A); the levels of circulating neutrophils, as well as monocytes, were significantly lower in the hRabMab1+gem/PTX cohort when compared to the gem/PTX cohort (*p* = 0.035 and *p* = 0.051, respectively).

Neutrophils and monocytes recruited from the circulation promote tumor progression in PDAC [20,21]. To address the potential physiological significance of neutrophil count normalization by hRabMab1+gem/PTX in the PaCa-44 model, we performed correlation analysis between circulating neutrophil counts and tumor volumes across all 4 cohorts; a highly significant positive correlation between neutrophil counts and tumor volumes was identified (R = 0.628, *p* = 0.00002), mirroring findings reported in human patients [22]. No differences in body weight of mice were observed between gem/PTX and hRabMab1+gem/PTX (not shown).

## 3. Discussion

In this study, we evaluated the effects of hRabMab1, administered intravenously as a single agent and in combination with gem/PTX, on primary tumor growth and systemic spread of orthotopically implanted human PDAC cells. Our main findings are as follows: (i) hRabMab1 was able to suppress liver metastases as a single agent while improving the efficacy of gem/PTX in suppressing metastases to other anatomical sites; (ii) hRabMab1 was able to penetrate primary tumor tissue when co-administered with gem/PTX, which led to the suppression of cancer cell proliferative potential, as well as leukocyte infiltration of primary tumors; (iii) the combination of hRabMab1+gem/PTX normalized WBC counts in tumor-bearing mice. These results agree with and expand on our published findings pointing to hRabMab1’s potential to stem the growth of human PDAC cells in vivo [15,23]; hRabMab1’s potential to increase the anti-metastatic efficacy of gem/PTX with no additional toxicity is particularly significant from a clinical perspective. The observations we describe here pose a number of new questions about the biologic function(s) of hRabMab1, e.g., what are the mechanisms underlying its ability to penetrate primary tumor tissue when co-administered with gem/PTX? Are anti-metastatic effects of hRabMab1 largely due to its ability to suppress the growth of PDAC cells already homed to distal sites, or is there also an effect on CTC intravasation, extravasation, and/or homing capacity? What causes the normalization of WBC counts in the hRabMab1+gem/PTX cohort? With regard to the tumor-tissue penetrance of hRabMab1, the most likely explanation is that the gem/PTX-elicited disruption of the fibrous capsule facilitated hRabMab1’s diffusion throughout the tumor ECM. With regard to the anti-metastatic properties of hRabMab1, these effects are most likely exerted on PDAC cells at various steps of metastatic dissemination. Integrins largely mediate a CTC’s capacity for motility and metastatic colonization, as well as their anchorage-independent survival, and ECM-β1 integrin interactions contribute to chemotherapy resistance of orthotopically grown PDAC tumors [24,25,26]. Thus, when considering hRabMab1′s perceived mode of action, i.e., the diminution of asTF-induced integrin activation, our data indicate hRabMab1 likely disrupts these integrin-driven metastatic processes. We note that the suppression of liver metastases by hRabMab1 alone may be due to the highly vascularized nature of the liver, which may have facilitated hRabMab1’s access to liver metastases. Further studies using additional primary tumor-driven endogenous metastatic models, as well as tail vein injections and hemi-splenic injections, will address whether hRabMab1′s suppresses PDAC cell seeding in the lung and the liver, respectively. Likewise, future studies are planned to feature tumor harvesting at regular intervals so that we can better assess the longitudinal dynamics of hRabMab1’s effects on primary tumor growth. Lastly, the normalization of WBC counts in the hRabMab1+gem/PTX cohort likely reflects a lowered systemic response due to a decreased cancer cell burden in these mice. It cannot be fully excluded, however, that the combination of hRabMab1+gem/PTX was more toxic than gem/PTX alone; that being said, such a scenario is unlikely given that there were no significant differences in BW between gem/PTX and hRabMab1+gem/PTX cohorts.

asTF is a soluble TF variant that has no role in normal blood clotting. flTF and asTF both play a role in cancer progression and interact with β1 integrins, yet with different consequences. In non-malignant cells, flTF keeps β1 integrins in an inactive state whereas in cancer, flTF activates αβ subsets and triggers protease-activated receptor 1 activation and thrombin generation that, collectively, promote cancer progression [23]. Earlier findings from other groups that the depletion of flTF can reduce tumor growth and thrombosis in murine models led to the exploration of targeting “total TF” in human clinical trials. The most well-characterized TF-targeting therapy, an antibody-drug conjugate (ADC) tisotumab vedotin (TF epitope unspecified), has shown promise with a phase II objective response rate of 15.6%; however, nearly 50% of patients had a treatment-emergent serious adverse event and nearly 70% experienced epistaxis [27,28,29]. While (fl)TF is still being actively pursued clinically, we posit that asTF is the preferred TF form to target in PDAC and other solid tumors due to both a low risk of bleeding complications, as well as superior selectivity for cancer cells and tissues; further, hRabMab1 is not an ADC and is, thus, not likely to cause tissue toxicity. As mentioned in the Introduction, “total TF” was recently implicated in the development of resistance to KRAS-G12C inhibitors [10]; as such, asTF-targeting via hRabMab1 may hold future promise in tackling this phenomenon in PDAC and other cancers driven by mutant KRAS.

Targeting asTF also comprises a novel way to impede integrin-linked, cancer-promoting signaling. By disrupting asTF-integrin interactions, hRabMab1 inhibits outside-in integrin signaling without the limitations associated with direct pharmacological inhibition of integrins: the key adverse issue of direct integrin inhibitors being their paradoxical ability to induce a conformational change in the integrin dimer, leading to a high-affinity ligand-binding state [30]. Indirectly inhibiting integrin-linked outside-in signaling cascades, however, has been one of the few methodologies that have provided additional benefit to mainline gemcitabine chemotherapy in the treatment of PDAC. When combined with gemcitabine and pembrolizumab, defactinib, a small molecule inhibitor of focal adhesion kinase (FAK), one of the kinases downstream of asTF/integrins in PDAC cells [14], showed preliminary efficacy without added toxicity [31]. We note that the use of antibody inhibitors of checkpoint proteins in combination with gemcitabine, with or without PTX, has largely yielded no additional benefit in the clinic in the treatment of PDAC, highlighting the importance of defactinib in this therapeutic regimen. Indeed, most monoclonal antibodies (mAbs) have failed in the clinic for the treatment of PDAC, largely due to the robust nature of PDAC tumor capsules, poor tumor vascularity, and the choice of therapeutic targets. Excitingly, we have recently seen that mAbs do have a place in the clinic for the treatment of PDAC, as the combination of pamrevlumab, a mAb targeting connective tissue growth factor (CTGF), with gem/PTX for the neoadjuvant treatment of locally advanced PDAC enabled study participants to advance to successful surgical resection in 8 of 24 cases (33%) compared to 1 of 13 (8%) patients who received gem/PTX alone [32]. Interestingly, CTGF elicits cellular mitogenic responses through an αvβ3-dependent mechanism in microvascular endothelial cells in vitro [33]. Thus, targeting integrin-linked signaling appears to hold promise as a new way to treat PDAC; we note that inhibiting entities upstream of integrin-linked signaling, such as asTF, is conceptually more robust compared to downstream targeting. We show here that when combined with gem/PTX, hRabMab1 both effectively penetrate tumor tissue and suppresses metastases.

In conclusion, we here show for the first time that asTF-inhibitory humanized antibody hRabMab1 holds promise to enhance the anti-metastatic effects of chemotherapy in PDAC. The main limitation of our study comprises the use of a single PDAC cell line PaCa-44; we note, however, that this cell line yields primary tumors that spontaneously metastasize to relevant anatomical sites, which augments the likely biological significance of the obtained results. Moreover, we previously showed the cancer cell-suppressing effect of inhibiting asTF in models that used other asTF-proficient PDAC cell lines. Future studies will explore hRabMab1’s ability to suppress primary tumor growth and experimental metastases in models featuring additional asTF-proficient PDAC cell lines and patient-derived xenografts.

## 4. Materials and Methods

### 4.1. PDAC Cell Lines and Culture Conditions; Western Blotting

Human PDAC cell lines Pt45.P1 (a kind gift of Prof. Holger Kalthoff), PaCa-44 (a kind gift of Prof. Stephan Haas), and HS766.T (ATCC) were cultured in DMEM supplemented with 10% fetal calf serum and antibiotics. Lysates were prepared with RIPA buffer (50 mM Tris HCl, pH 8.0; 150 mM NaCl, 0.5% *w*/*v* sodium deoxycholate; 0.1% SDS; 1% NP-40) containing 5mM EDTA along with Halt protease and phosphatase inhibitor cocktail (Thermo Fisher Scientific, Waltham, MA, USA, ref. 1861281) and loaded into 10% TGX gels (BioRad Hercules, CA, USA). Protein was transferred onto PVDF membranes and probed with antibodies specific for human α6 integrin (Cell Signaling Technology, Danvers, MA, USA, #3750, 1:1000), β1 integrin (Cell Signaling Technology, Danvers, MA, USA, #34971, 1:1000), flTF (clone TF9-10H10, Invitrogen/Thermo Fisher Scientific, Waltham, MA, USA, 1:500), asTF (custom rabbit polyclonal, ref. 14, 2 μg/mL), and beta-actin (Cell Signaling Technology, Danvers, MA, USA, #3700, 1:1000).

### 4.2. In Vivo Studies

Orthotopic tumor implantation: On the day of surgery, PaCa-44 cells were detached from tissue culture plates with 0.25% trypsin. Trypsin was neutralized with DMEM containing 10% FBS. Cells were washed 2× with DMEM before preparing a final suspension of 2.5 × 10^7^ cells/mL. Then, 20 µL of PaCa-44 cell suspension (containing 5 × 10^5^ cells) was injected into the pancreata proximal to the duodenum of NOD.scid mice (The Jackson Laboratory, Bar Harbor, ME, USA, 001303). Ten days post-implantation, mice were randomized into four cohorts: vehicle; hRabMab1 (IV at 18 mg/kg); gem/PTX (50 mg/kg IP and 3 mg/kg IV, respectively); and hRabMab1+gem/PTX. In the vehicle and hRabMab1 cohorts, mice were sacrificed when tumor volume reached 1500 mm^3^; in the gem/PTX and hRabMab1+gem/PTX cohorts, mice were sacrificed on day 40 post-implantation. In vivo imaging was carried out weekly using the IVIS Spectrum System (Xenogen Corporation, Alameda, CA, USA). Blood counts were determined at sacrifice using a HEMAVET automated hematology analyzer (Drew Scientific, Miami Lakes, FL, USA); historical reference values for white blood cell count data for healthy NOD.scid mice were retrieved from Charles River hematology records [34]. Tumor volumes were derived using the formula V = (W(2) × L)/2 for caliper measurements. A portion of each tumor material was flash frozen, as well as fixed in 10% formalin and embedded in paraffin for RNA-seq and IHC analyses, respectively.

### 4.3. Immunohistochemistry

Formalin-fixed tissues were embedded in paraffin and sectioned into 5 μm sections. Sections were deparaffinized and rehydrated into PBS. Antigen retrieval was carried out whenever indicated by the antibody manufacturer; native peroxidase activity was squelched using 0.4% hydrogen peroxide. Blocking was carried out using 5% bovine serum albumin in PBS and sections were incubated with the following antibodies at the manufacturer-indicated dilutions: CD31 (Novus Biologicals, Centennial, CO, USA, AF3628, 10 µg/mL), CD206 (Cell Signaling Technology, Danvers, MA, USA, #24595, 1:200), Ki67 (Novus Biologicals, Centennial, CO, USA, NB110-89717, 1:250), Myeloperoxidase (Abcam Waltham, MA, USA, AB 300650, 1:1000), and goat anti-human IgG biotinylated antibody (Vector Laboratories, Newark, CA, USA, BA-3000-1.5, 1:500). Species-specific, HRP-conjugated anti-antibody polymers and DAB+ reagent (both—Cell Signaling) were used to visualize unlabeled primary antibody binding and HRP-streptavidin reagent (SA-5704, Vector Laboratories) was used to visualize anti-human IgG antibody; all sections were counterstained with hematoxylin. Representative images (n = 6 per tissue specimen) were captured using an Olympus BX51 (Center Valley, PA, USA) equipped with Olympus DP72 digital camera and used for statistical analyses. Staining intensity and/or positive staining events were analyzed using ImageJ.

### 4.4. RNA-seq

Using an RNeasy kit (Qiagen, Germantown, MD, USA), total RNA was isolated from frozen tissue specimens representing median tumor volumes from each experimental cohort. Directional polyA RNA-seq was performed by the Genomics, Epigenomics and Sequencing Core at the University of Cincinnati, using established protocols. The quality of total RNA was QC analyzed by Bioanalyzer (Agilent, Santa Clara, CA, USA). To enrich polyA RNA for library preparation, NEBNext Poly(A) mRNA Magnetic Isolation Module (New England BioLabs, Ipswich, MA, USA) was used with 1 µg good quality total RNA as input. Next, NEBNext Ultra II Directional RNA Library Prep kit (New England BioLabs) was used for library preparation under PCR (cycle number: 8). After library QC and Qubit quantification (ThermoFisher, Waltham, MA, USA), the normalized libraries were sequenced using NextSeq 2000 Sequencer (Illumina, San Diego, CA, USA) under the setting of PE 2x61 bp to generate an average of 42.3 M reads. Once the sequencing was completed, FASTQ files for downstream data analysis were generated and transferred/shared via BaseSpace Sequence Hub (Illumina). A quality control check on the FASTQ files was performed using FASTQC (https://www.bioinformatics.babraham.ac.uk/projects/fastqc/, accessed 21 December 2023) and MultiQC [35] to verify data quality. The FASTQ files were processed with STAR (v. 2.7.7a) [36] with the Gencode GRCh38 as an index to determine alignment. Salmon [37] quant (version 1.9.0), with default parameters and the RefSeq CRCm39 index, was used to obtain counts. Differential gene expression analysis was performed with unnormalized counts and default parameters using the DESeq2 package (Ver 1.34.0) between the following groups of samples: vehicle vs. hRabMab1, vehicle vs. gem/PTX, vehicle vs. hRabMab1+gem/PTX [38,39]. Wald test was used to test the null hypothesis of no differential expression across the two sample groups along with Benjamini–Hochberg correction to adjust for multiple testing. Functional gene set enrichment was performed with lists of genes that had significant differential expression (adjusted *p*-value < 0.05) using the ToppFun application in the ToppGene suite (https://toppgene.cchmc.org/) [40].

### 4.5. Statistics

Continuous variables were summarized as means ± SD or median and inter-quartile ranges (IQR). Two-tailed t test and 1-way ANOVA with Tukey’s multiple comparison test were used to test the difference between two or more cohorts, respectively (GraphPad Prism v.6.0); *p* values ≤ 0.05 were deemed significant. Categorical variables were summarized as counts and percentages. The Chi-square test was used to test the association.

## Figures and Tables

**Figure 1 ijms-25-02580-f001:**
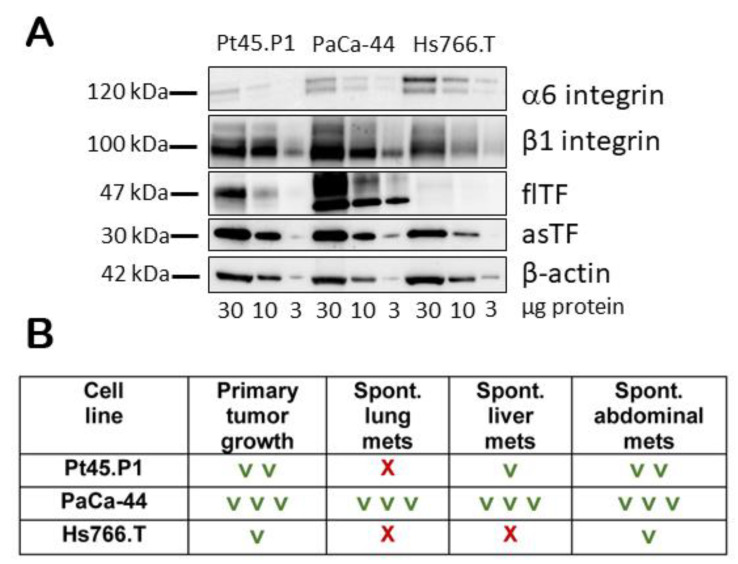
(**A**) Expression of TF protein variants and α6β1 integrins in Pt45.P1, PaCa-44, and Hs766.T cells. (**B**) In vivo properties of Pt45.P1, PaCa-44, and Hs766.T cells were evaluated (2 in vivo studies per cell line); **∨**: positive outcome; **X**: negative outcome.

**Figure 2 ijms-25-02580-f002:**
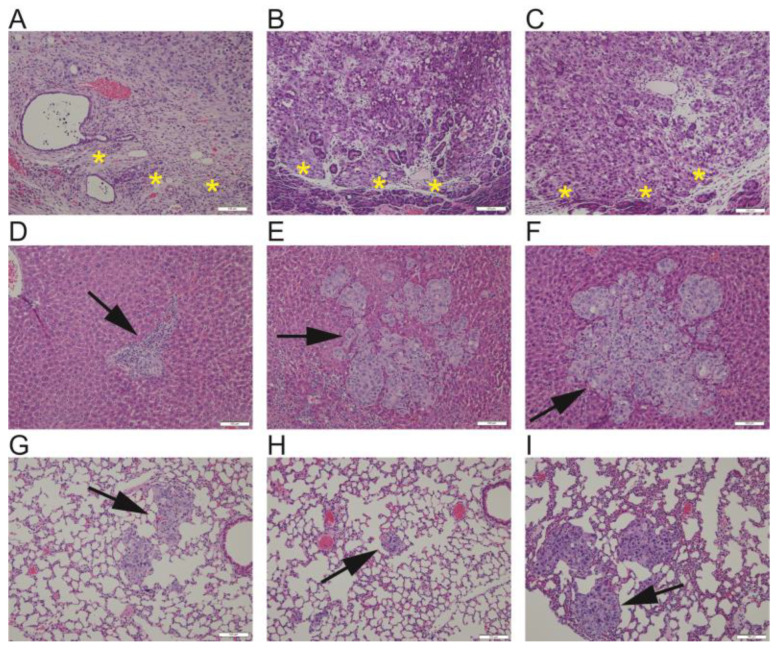
Representative images of PaCa-44 primary tumor leading edge (demarked by yellow asterisks, (**A**–**C**)), liver metastases (arrows, (**D**–**F**)), and lung metastases (arrows, (**G**–**I**)) from 3 different mice; animal-specific tissues arranged in vertical columns. Hematoxylin and eosin stain; original magnification: 20×; 100 µm scale bar shown in bottom right of each micrograph.

**Figure 3 ijms-25-02580-f003:**
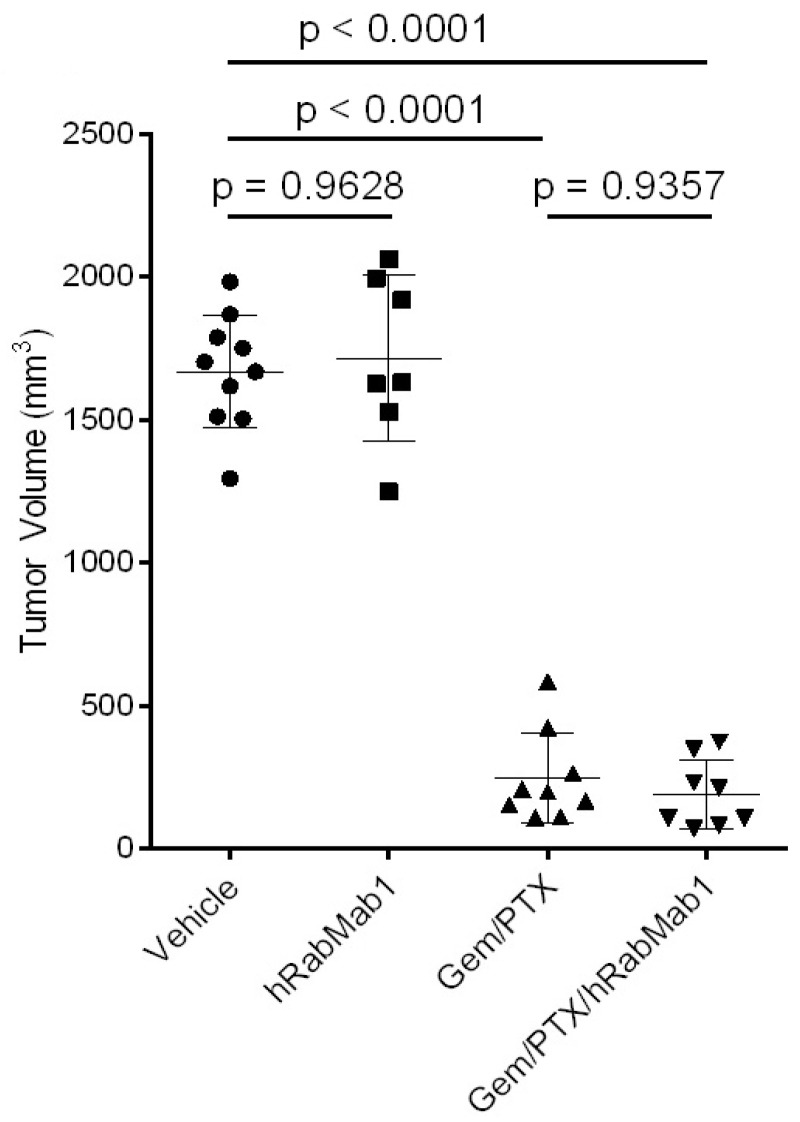
Tumor volume in four experimental cohorts as indicated; 1-way ANOVA with Tukey’s multiple comparison test was used to assess significance.

**Figure 4 ijms-25-02580-f004:**
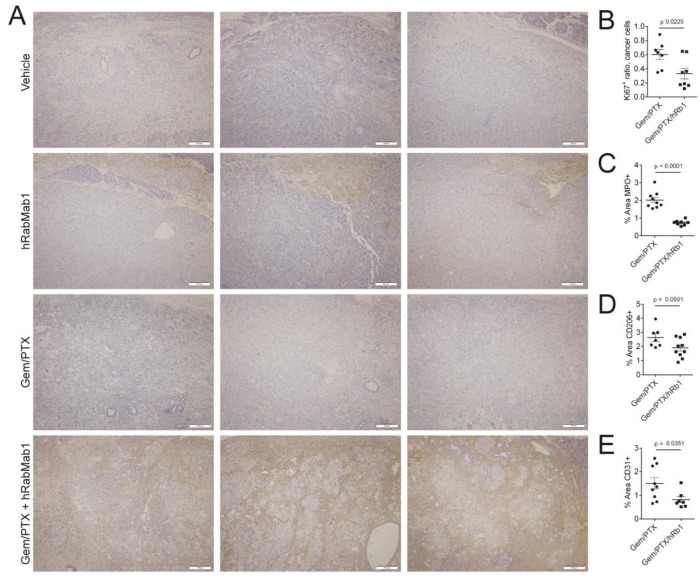
(**A**) IHC of PaCa-44 primary tumor tissue (3 representative specimens per cohort as indicated) stained for human IgG; original magnification: 20×; 100 µm scale bar shown in bottom right of each micrograph. (**B**–**E**) Quantification of Ki67 positivity; neutrophil infiltration; monocyte/macrophage infiltration; and neovascularization, respectively; please see Section 4 for details.

**Figure 5 ijms-25-02580-f005:**
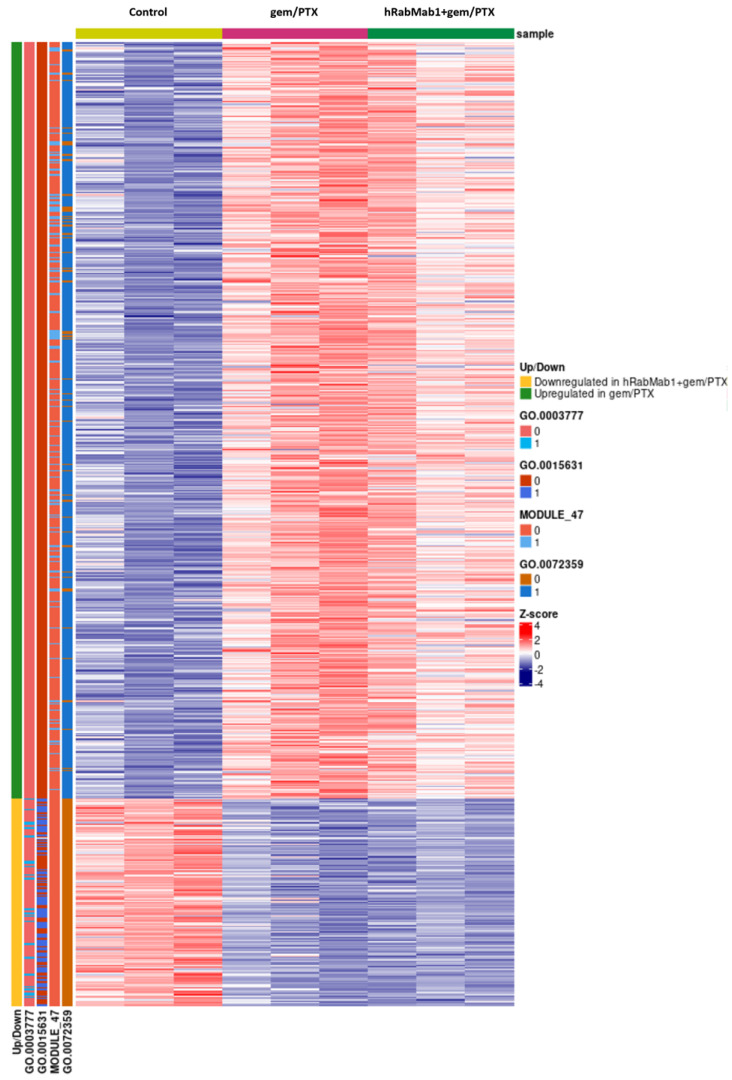
Heatmap of z-score normalized, variance stabilizing transformed differentially expressed gene counts. Up/Down trackbar indicates whether the gene is upregulated in gem/PTX samples, or downregulated in hRabMab1+gem/PTX samples. The trackbars for Gene Ontology terms indicate the presence (1) or absence (0) of the gene in the set. GO:0003777: microtubule motor activity; GO:0015631: tubulin binding; GO:0072359: —circulatory system development; MODULE_47: ECM and collagens.

**Figure 6 ijms-25-02580-f006:**
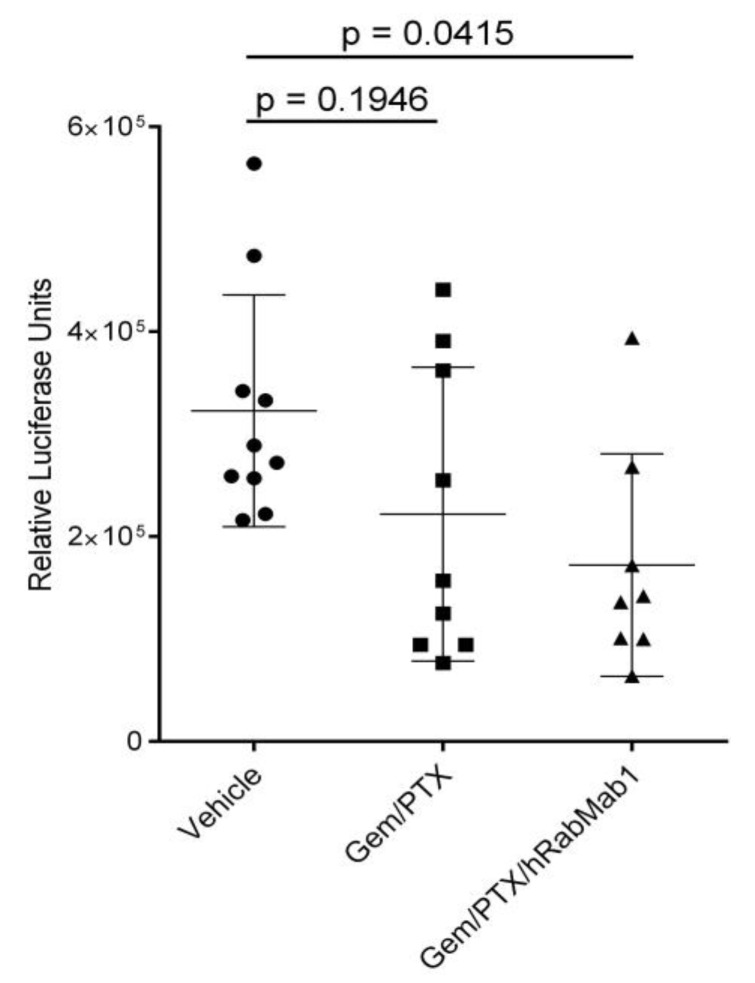
Cumulative metastatic spread assessed by quantitative luciferase imaging in experimental cohorts as indicated; 1-way ANOVA with Tukey’s multiple comparison test was used to assess significance.

**Figure 7 ijms-25-02580-f007:**
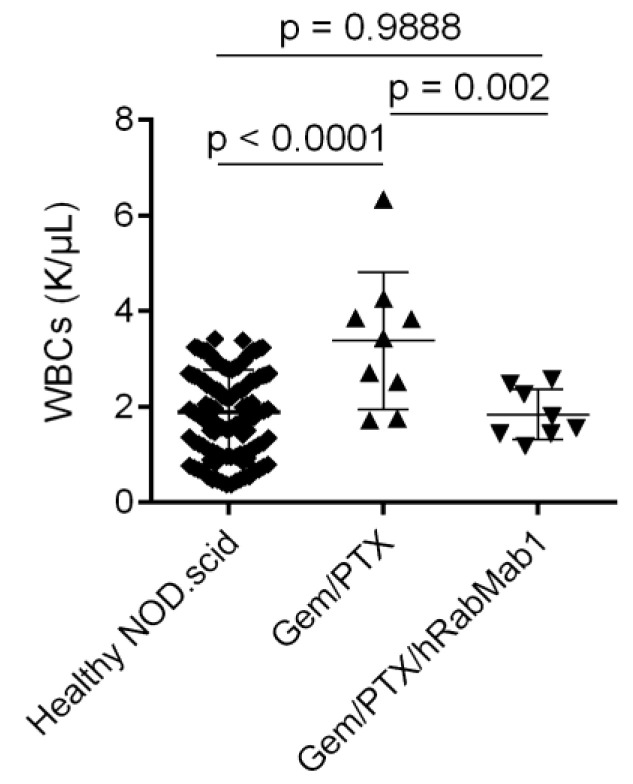
White blood cell (WBC) counts in NOD.scid mice: historical NOD.scid reference data and the experimental cohorts as indicated; 1-way ANOVA with Tukey’s multiple comparison test was used to assess significance.

## Data Availability

The datasets presented in this study can be found in the online repository; accession number: GSE252286.

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
