# Peer review of "First-in-Class Humanized Antibody against Alternatively Spliced Tissue Factor Augments Anti-Metastatic Efficacy of Chemotherapy in a Preclinical Model of Pancreatic Ductal Adenocarcinoma"

_ijms, 2024, doi:10.3390/ijms25052580_

Round 1

Reviewer 1 Report

Comments and Suggestions for Authors

This article studies the effect of humanized antibody against alternatively spliced tissue factor augments anti-metastatic efficacy of chemotherapy in pancreatic ductal adenocarcinoma, which is a valuable article. But there are a few points that need to be strengthened:

1. Some of the resolution of the Wester blot in Figure 1 is not good enough, and better images need to be provided.

2. In the introduction, the relationship between TF and PDCA can be more emphasized and explained.

3. The animal experiment approval number from IACUC must be written into the method.

4. Regarding the number of total WBC in the experiment, is there any individual data for white blood cell classification?

5. In the discussion, the future application value of this article can be more emphasized.

Comments on the Quality of English Language

No

Author Response

REVIEWER 1

This article studies the effect of humanized antibody against alternatively spliced tissue factor augments anti-metastatic efficacy of chemotherapy in pancreatic ductal adenocarcinoma, which is a valuable article.

Response: we thank Reviewer 1 for their careful and helpful evaluation of our work.

 But there are a few points that need to be strengthened:

1. Some of the resolution of the Western blot in Figure 1 is not good enough, and better images need to be provided.

Response: as requested by Reviewer 1, we replaced the Fig.1 western blot file with a higher-resolution version.

2. In the introduction, the relationship between TF and PDAC can be more emphasized and explained.

Response: done as requested; references have been newly added to expand on TF in PDAC.

3. The animal experiment approval number from IACUC must be written into the method.

Response: we thank Reviewer 1 for alerting us to the absence of this information. Our IACUC-approved protocol number, 21-09-16-01, is now included in the manuscript.

4. Regarding the number of total WBC in the experiment, is there any individual data for white blood cell classification?

Response: the data describing WBC subsets is now included (Supplementary Figure 3).

5. In the discussion, the future application value of this article can be more emphasized.

Response: done as requested; references have been newly added to expand on the future application value of our first-in-class biologic.

Reviewer 2 Report

Comments and Suggestions for Authors

Comments:

This is a novel study by Lewis et al. Here, they have stated that this is the first study demonstrating that hRabMab1 may help suppress metastasis in PDAC. Their study describes RabMab1’s ability to improve the efficacy of chemotherapy, which is particularly significant and warrants further investigation. A careful review of this manuscript brings up some major concerns that should be addressed to improve the quality of the manuscript. The comments are listed below.

1.    Figure 1 should be divided into A and B.

2.    The authors indicate that TF-proficient, KRAS G12V-mutant cell line PaCa44 yields reproducible-size, aggressive primary orthotopic tumors that spread spontaneously to PDAC-relevant anatomical sites (Figure 2).

 3.    The images require scale bars to confirm that these images are comparable in magnification. Figure 2 shows lung metastases (arrows, G-I) from 3 different mice; animal-specific tissues arranged in vertical columns appeared different in the context of their areas. A scale bar will be helpful here.

 4.    The results in Figure 3 are precise, but additional resected tumors might be helpful to understand the natural time effect of the treatment on the tumor volume. Moreover, the analysis didn’t show the outcome from the hRabMab1 and did not significantly impact gem/PTX in combination with gem/PTX. Hence, the author’s claim is not fulfilled (Figure 3).

 5.    Again, figure 4 should be accompanied by a scale bar.

 6.    The results in Figure 4 is confusing. Ki67+ signal was significantly suppressed by the addition of hRabMab1 to gem/PTX; there were also considerably fewer neutrophils, monocytes, and microvessels in hRabMab1+gem/PTX tumors compared to gem/PTX tumors (Figure 4B-E). How were the Immune cells quantified in Fig 4 B-E? Here, representative images for the quantifications will be helpful.

 7.    In Figure 6 metastatic spread (liver, lung, abdominal cavity) was not significantly reduced by gem/PTX alone; however, the addition of hRabMab1 to gem/PTX significantly reduced whole-body metastatic spread (p=0.0415, vehicle vs hRabMab1+gem/PTX, Figure 6). Here, including the analysis in the hRabMab1-only group will be helpful.

 8.    The authors state that “In the liver, hRabMab1 significantly reduced metastatic burden as a single agent to a comparable degree of that achieved by gem/PTX (hRabMab1: p=0.0089 vs vehicle; gem/PTX: p=0.0008 vs. vehicle). Are there any results to support this finding? Please shed some light on the comparison between the hRabMab1 vs gem/PTX, which is missing from the discussion.

 Some minor comments include:

1.    Detailed cell culture methods would be helpful to recapture the results in different settings.

2.    Please expand the lysate preparation method. Whether phosphatase or PICs were used?

3.  Please confirm whether the gem/PTX administered the IV route.

Author Response

REVIEWER 2

This is a novel study by Lewis et al. Here, they have stated that this is the first study demonstrating that hRabMab1 may help suppress metastasis in PDAC. Their study describes RabMab1’s ability to improve the efficacy of chemotherapy, which is particularly significant and warrants further investigation.

Response: we thank Reviewer 2 for their careful and helpful evaluation of our work.

A careful review of this manuscript brings up some major concerns that should be addressed to improve the quality of the manuscript. The comments are listed below.

1. Figure 1 should be divided into A and B.

Response: done as requested.

2. The authors indicate that TF-proficient, KRAS G12V-mutant cell line PaCa44 yields reproducible-size, aggressive primary orthotopic tumors that spread spontaneously to PDAC-relevant anatomical sites (Figure 2).

Response: that is correct; a reference was newly added with regard to PaCa-44 cell line mutation status.

3. The images require scale bars to confirm that these images are comparable in magnification. Figure 2 shows lung metastases (arrows, G-I) from 3 different mice; animal-specific tissues arranged in vertical columns appeared different in the context of their areas. A scale bar will be helpful here.

Response: done as requested.

4. The results in Figure 3 are precise, but additional resected tumors might be helpful to understand the natural time effect of the treatment on the tumor volume. Moreover, the analysis didn’t show the outcome from the hRabMab1 and did not significantly impact gem/PTX in combination with gem/PTX. Hence, the author’s claim is not fulfilled (Figure 3).

Response: we appreciate Reviewer 2’s contention that longitudinal measurements of tumor volume dynamics would comprise valuable information regarding hRabMab1’s effects. We will be addressing this in future studies and added a sentence to the discussion in this regard. Because the text associated with Fig.3 reads as follows: “Post-mortem analysis of primary tumors revealed that hRabMab1 did not significantly impact the primary tumor volume when used alone or in combination with gem/PTX,” we believe that we correctly described our observations regarding the endpoint tumor volumes in our cohorts.

5. Again, figure 4 should be accompanied by a scale bar.

Response: done as requested.

6. The results in Figure 4 is confusing. Ki67+ signal was significantly suppressed by the addition of hRabMab1 to gem/PTX; there were also considerably fewer neutrophils, monocytes, and microvessels in hRabMab1+gem/PTX tumors compared to gem/PTX tumors (Figure 4B-E). How were the Immune cells quantified in Fig 4 B-E? Here, representative images for the quantifications will be helpful.

Response: done as requested; representative images are now included in the supplement (Supplementary Figure 1). Methods section 4.3 describes the quantification methodology as follows: “Representative images (n=6 per each tissue specimen) were captured using Olympus BX51 equipped with Olympus DP72 digital camera and used for statistical analyses. Staining intensity and/or positive staining events were analyzed using ImageJ.”

7. In Figure 6 metastatic spread (liver, lung, abdominal cavity) was not significantly reduced by gem/PTX alone; however, the addition of hRabMab1 to gem/PTX significantly reduced whole-body metastatic spread (p=0.0415, vehicle vs hRabMab1+gem/PTX, Figure 6). Here, including the analysis in the hRabMab1-only group will be helpful.

8. The authors state that “In the liver, hRabMab1 significantly reduced metastatic burden as a single agent to a comparable degree of that achieved by gem/PTX (hRabMab1: p=0.0089 vs vehicle; gem/PTX: p=0.0008 vs. vehicle). Are there any results to support this finding? Please shed some light on the comparison between the hRabMab1 vs gem/PTX, which is missing from the discussion.

Response: we appreciate Reviewer 2’s points 7 and 8, which focus on our finding that the single-agent effect of hRabMab1 was limited to suppressing liver metastases. Because hRabMab1 was not able to penetrate tumor tissue unless combined with gem/PTX (Fig.3), we elected to devote the final results section 2.3 and its accompanying Figs. 6 and 7 on the systemic effects of hRabMab1 in combination with gem/PTX; we thus believe that the manner in which we present the data in Figs. 6 and 7 is appropriate. However, we do agree that the single-agent effect of hRabMab1 on liver metastases is interesting and can thus be described in more detail; as such, we added a new figure to this effect (Supplementary Figure 2).

 Some minor comments include:

1. Detailed cell culture methods would be helpful to recapture the results in different settings.

Response: we fully agree that adding such detail is helpful to the reader. Only routine cell culture conditions were utilized. As indicated, our human PDAC cell lines were cultured in DMEM supplemented with 10% FBS and antibiotics. Details of cell handling procedures on the day of surgery have been added to the methods: “On the day of surgery, PaCa-44 cells were detached from tissue culture plates with 0.25% trypsin. Trypsin was neutralized with DMEM containing 10% FBS. Cells were washed 2x with DMEM before preparing a final suspension of 2.5x107 cells/mL. 20 µL of PaCa44 cell suspension (containing 5x105 cells) was injected into the pancreata proximal to the duodenum of NOD.scid mice (Jackson 001303).”

2. Please expand the lysate preparation method. Whether phosphatase or PICs were used?

Response: we thank Reviewer 2 for noticing this oversight. This information is now included.

3. Please confirm whether the gem/PTX administered the IV route.

Response: we thank Reviewer 2 for [pointing out this ambiguity. We have clarified in the methods that gem was administered IP and PTX administered IV.